# Design and Implementation of an LSTM Model with Embeddings on MCUs for Prediction of Meteorological Variables

**DOI:** 10.3390/s25123601

**Published:** 2025-06-07

**Authors:** Jhan Piero Paulo Merma Yucra, David Juan Cerezo Quina, German Alberto Echaiz Espinoza, Manuel Alejandro Valderrama Solis, Daniel Domingo Yanyachi Aco Cardenas, Andrés Ortiz Salazar

**Affiliations:** 1Professional School of Electronic Engineering, Universidad Nacional de San Agustín de Arequipa, Arequipa 04002, Peru; jmermay@unsa.edu.pe (J.P.P.M.Y.); dcerezo@unsa.edu.pe (D.J.C.Q.); dyanyachi@unsa.edu.pe (D.D.Y.A.C.); 2Professional School of Telecommunications Engineering, Universidad Nacional de San Agustín de Arequipa, Arequipa 04002, Peru; mvalderramasol@unsa.edu.pe; 3Department of Computer Engineering and Automation, Federal University of Rio Grande do Norte (DCA-UFRN), Natal 59072-970, RN, Brazil; andres@dca.ufrn.br

**Keywords:** edge computing, embeddings, LSTM, climate prediction

## Abstract

The use of recurrent neural networks has proven effective in time series prediction tasks such as weather. However, their use in resource-limited systems such as MCUs presents difficulties in terms of both size and stability with longer prediction windows. In this context, we propose a variant of the LSTM model, which we call SE-LSTM (Single Embedding LSTM), which uses embedding techniques to vectorially represent seasonality and latent patterns through variables such as temperature and humidity. The proposal is systematically compared in two parts: The first compares it against other reference architectures such as CNN-LSTM, TCN, LMU, and TPA-LSTM. The second stage, which includes implementation, compares it against the CNN-LSTM, LSTM, and TCN networks. Metrics such as the MAE and MSE are used along with the network weight, a crucial aspect for MCUs such as an ESP32 or Raspberry Pi Pico. An analysis of the memory usage, energy consumption, and generalization across different regions is also included. The results show that the use of embedding optimizes the network space without sacrificing the performance, which is crucial for edge computing. This research is part of a larger project, which focuses on improving agricultural monitoring systems.

## 1. Introduction

Weather prediction using recurrent neural networks such as Long Short-Term Memory (LSTM) has proven to be effective in learning temporal dependencies [1]; however, its implementation in systems such as Microcontroller Units (MCUs) presents some challenges. These problems include the model size and the performance stability when using a longer prediction window time [2]. Although solutions such as TPA-LSTM [3] or Legendre Memory Unit (LMU) [4] improve the accuracy, their complexity makes them unfeasible for these devices. In contrast, the use of AI and the IoT can solve this problem [5,6]. Other architectures, such as CNN-LSTM [7] and TCN [8], can be implemented in MCUs and are considered in this study for comparative purposes. This research prioritizes a completely local solution, performing inferences on an Microcontroller Unit (MCU). This allows us to utilize its advantages, such as privacy, by avoiding data transfer to the cloud, achieving minimal latency due to an on-premises execution, attaining computational resource efficiency, and maintaining accessibility in environments with limited connectivity, as demonstrated in comparative studies of distributed architectures [9] and in model optimization solutions for constrained hardware [10]. This approach allows for more autonomous and efficient solutions in environments with low connectivity or a lack of a robust infrastructure. By performing processing locally, the dependence on third parties is minimized, and, in addition, the low energy consumption of MCUs is leveraged.

In this article, we present an LSTM model with embedding layers, a technique that, as demonstrated in [11], allows the size of the neural network to be reduced without sacrificing its performance. This technique is similar to how embeddings are used in natural language processing (NLP) to represent semantic relationships between words [12]. For this application, embeddings are learned during the training phase as vectors, which find the relationships between seasons based on the temperature and humidity series. In addition, to explicitly capture seasonality, a label corresponding to the season was designated, grouping the months into quarters according to the meteorological convention and taking into account the hemisphere where each meteorological station is located. In the southern hemisphere, the distribution is as follows: December–February (summer), March–May (autumn), June–August (winter), September–November (spring); in the northern hemisphere, the distribution is reversed: the grouping is reversed: December–February (winter), March–May (spring), June–August (summer), September–November (autumn) [13]. This category was encoded using embeddings, allowing the neural network to learn latent representations of seasonality and its influence on climatic variables, such as temperature and humidity. Similar to their use in sequential tasks, such as time series or natural language processing, the categorical variables are represented using embedding vectors that capture relationships and patterns over time. Once the network completes the training, these vectors are integrated as input to the LSTM. Thus, the network reduces the number of parameters and the final model size while maintaining the quality or accuracy of a larger architecture.

We performed three main comparisons, all evaluating the MAE, MSE, RMSE, MAPE, RSE, and R^2^ metrics in the time windows t + 1, t + 3, and t + 6. All of the results shown in the tables are normalized to facilitate the comparison between models. The first comparison measured how much the size is reduced without losing performance, similarly to other networks such as LMU, TPA-LSTM, CNN-LSTM, TCN, and the standard LSTM. The second comparison focused on the implementation of these models on various MCU boards, also considering the inference time and power consumption, which are important aspects in real-world applications [14]. For the compatibility reasons explained in the results and implementation sections, TPA-LSTM and LMU were not included. Finally, the third comparison consisted of validating the effect of using embeddings in the LSTM, contrasting the base model with the proposed SE-LSTM model that integrates these vectors. This comparison was conducted in 13 cities at different latitudes.

The model maintains consistency in predictions when run on different hardware platforms, validating its portability. All resources (dataset, code, and trained models) are available as indicated in the Data Availability Statement to facilitate their adoption in practical applications.

## 2. Materials and Methods

This study used temperature and humidity data obtained from NASA’s POWER [15], with coordinates of latitude −16.4333 and longitude −71.5617, covering the periods from January 2020 to March 2025, with a sampling frequency of one hour, yielding a total of 45,864 data points. These data points were distributed as follows: 80% for training, 15% for validation, and 5% for testing. Temperature and humidity were selected as the main variables due to their high availability and temporal continuity in different geographic locations, allowing for a comparison of the model performance in different contexts. Furthermore, missing values marked as −999 are very rare in these variables, which facilitated their processing and maintained temporal continuity without affecting the model. Other meteorological variables, such as solar radiation and wind speed, were excluded due to their lower availability and continuity in the datasets used. Each sample contains the date and time, as well as the temperature and humidity values, which were normalized between 0 and 1. Discordant data (values −999) caused by missing data on certain dates were eliminated, as indicated by POWER NASA. In cases where the value −999 is not repeated more than twice consecutively, the missing data were replaced by the average of the immediately adjacent values in order to preserve the continuity of the time series without introducing distortions.

Because both temperature and humidity are related to the climatic seasons, the seasons were coded as an integer (0−3) and incorporated using the four-dimensional embedding layer. The coding process groups the months into four seasons according to meteorological convention: December–February (0), March–May (1), June–August (2), September–November (3). This layer initializes its vectors randomly and, during training, adjusts these vectors by backpropagation of the error—along with the rest of the network weights—to minimize the loss function. In this way, the model automatically learns to locate stations with similar weather patterns close to each other in the vector space, improving the LSTM’s ability to capture temporal and seasonal relationships. The optimal number of dimensions was determined by modifying the parameters of the embedding layer and evaluating the model’s overall performance.

As illustrated in Figure 1, the relationships between the seasons are represented by by vectors in the 2D and 3D planes:

The arrangement of points on a semicircle illustrates how the embedding vectors captured seasonal variations in the climate variables. Using a PCA dimensionality reduction, these four-dimensional vectors allow for the connections between seasons. Contrary seasons, such as summer and winter, are often reflected in opposite areas of the plane, as the embeddings have learned to represent their different weather patterns (such as high temperatures and low humidity under opposite circumstances). Spring and autumn, during periods of greater climatic stability, emerge in the transitional areas. This illustration shows how the model was able to gather seasons with similar climatic behaviors close to each other in the vector space.

The architecture is structured as an LSTM network organized into three levels. The initial layer has 64 units and uses the return_sequences function to collect temporal data for the entire sequence. The second layer has 32 units and also uses the return_sequences function, and the third LSTM layer is composed of 8 units. All of the layers use the ReLU activation function, which provides nonlinearity in the variables. The output layer is a dense layer with 6 units and a linear activation, for the 3 temperature and 3 humidity predictions, corresponding to the time windows t+1h, t+3h, and t+6h. Thus, the input to the model is 24 samples of both temperature and humidity, concatenated to the 4-dimensional embedding layer learned during training.

The following flowchart, shown in Figure 2, illustrates how station embeddings are integrated with the LSTM network.

The integration of embeddings with the LSTM network allows the model to learn the context of the seasons and the relationship between temperature and humidity over the following hours.

The model was trained for 115 epochs, using the Adam optimizer, along with a learning rate of 0.001, a batch size of 64, and a loss function of the MSE. The hyperparameters were progressively defined, taking into account the behavior of the performance metrics in the test set, with the goal of achieving good performance.

Once the training is complete, the model is able to predict future temperature and humidity levels in the aforementioned time windows; at the same time, the learned vectors are also obtained, and in each training, these can change in magnitude, since they may give more weight to a certain variable during a certain season.

## 3. Results

This section is divided into four main parts. The design stage compares the performance of the proposed model, SE-LSTM, against other neural networks in the aforementioned city—networks such as CNN-LSTM, TCN, TPA-LSTM, LMU, and standard LSTM. In the second part, the dynamics of the embeddings are obtained. The third part compares the performance in various MCUs, and the fourth part involves the validation in various cities around the world other than the city in the first part.

### 3.1. Comparison at the Design Stage

The models compared were CNN-LSTM, TCN, LSTM, TPA-LSTM, LMU, and the proposed SE-LSTM model. The performance metrics used to compare were the MAE, MSE, RMSE, MAPE, RSE, $R^{2}$, and the trained network weight. They were evaluated over three time horizons: t + 1 h, t + 3h, and t + 6 h, where “t” is the time after the collection of the previous 24 samples. An analysis of the variation in the metrics in each window and a summary of the overall performance are presented. The images accompanying the tables correspond to the model’s predictions of the test data. All of the metrics were calculated on the test set and applied to all of the networks analyzed.

#### 3.1.1. First Window (t + 1)

In the first time window, the TPA-LSTM model shows the lowest errors (MSE, MAE, RMSE) and a very high $R^{2}$, indicating its accuracy in nowcasting. 

The SE-LSTM model also performs well, with competitive metrics and an $R^{2}$ for temperature of 0.9781, demonstrating that the use of embeddings based on meteorological representations helps capture short-term patterns. The SE-LSTM predictions for temperature and humidity are illustrated in Figure 3 and Figure 4, respectively, while the performance metrics are summarized in Table 1.

#### 3.1.2. Second Window (t + 3)

In this window, the errors tend to increase for all of the models. SE-LSTM remains consistent, while the other models show greater performance degradation. This demonstrates that the use of embedding in SE-LSTM helps maintain stability in medium-term predictions, particularly for the temperature variable. The SE-LSTM predictions for temperature and humidity are illustrated in Figure 5 and Figure 6, respectively, while the performance metrics are summarized in Table 2.

#### 3.1.3. Third Window (t + 6)

As the prediction window grows, models like TPA-LSTM and TCN offer a good balance between accuracy and generalization, while LSTM and CNN-LSTM show a more noticeable performance drop. Overall, SE-LSTM establishes itself as the model most resilient to degradation in environments with high temporal uncertainty. The SE-LSTM predictions for temperature and humidity are illustrated in Figure 7 and Figure 8, respectively, while the performance metrics are summarized in Table 3.

As seen in Table 4, the use of embeddings in SE-LSTM is aimed at maintaining an optimal balance between accuracy and efficiency, unlike TPA-LSTM and LSTM, which seek to maximize performance at the expense of greater complexity (more neurons or layers). Overall, SE-LSTM slightly outperforms LSTM with lower weights and LMU with similar weights, proving to be more suitable for MCU applications. On the other hand, models such as TCN and CNN-LSTM show a competitive performance but with an intermediate weight.

As seen in the Table 1, Table 2, Table 3 and Table 4 the use of embeddings in SE-LSTM is aimed at maintaining an optimal balance between accuracy and efficiency, while TPA-LSTM and LSTM are aimed at achieving the highest possible performance, having more complex architectures or a greater number of neurons (the reasons for their weighting); as a general rule, SE-LSTM performs slightly better than LSTM at lower weights, and better than LMU at similar weights, for this specific application.

### 3.2. Learning Dynamics of the Embeddings

In the methodology section, it was explained that each spatial cell has an embedding vector that is adjusted during the training. To see if they are actually learning anything useful, we visualize how these vectors change over the epochs. As seen in Figure 9, they take shape from epoch 0 to 80. This allows us to observe how the model organizes its internal representation space as it trains.

### 3.3. Comparison in MCUs

The LMU and TPA-LSTM networks are not available in TFLite due to the use of advanced functions; therefore, in this section, we compare the performance of the CNN-LSTM, LSTM, TCN, and SE-LSTM networks. The embedding function is not available in TFLite, but we can emulate it by adding the matrices (generated by learning the relationships between temperature and humidity during training) as an additional input. These vectors often vary in magnitude; for example, in a given training session, the following were obtained, as shown in Table 5.

The implementation would then be as follows: 6 features composed of temperature, humidity, and a season (a 4-dimensional vector), with steps of 24, briefly (none, 24, 6), and finally flattened to an input of type (none, 144).

The architecture was also modified to 1 LSTM layer with the ReLU activation function, followed by two dense layers with 128 and 36 neurons, respectively, both with the ReLU activation, and an output layer with 6 neurons and a linear activation function for the prediction times per variable. The batch size was 32, with 80 epochs, and the remaining metrics remained the same. This configuration was applied to both the SE-LSTM and LSTM models. As can be seen in Table 6, the performance when using embeddings increases when the network weight is maintained.

#### 3.3.1. Edge Impulse

To upload the model to a board, several environments can be used, but the one we chose was the Edge Impulse platform [16] due to its ease of inserting the necessary libraries into the same downloadable.zip; for the compilation, TensorFlow Lite was used, and as shown in Table 7, the inference time is around 55 ms on an ESP32-S3, 671 ms on a Raspberry Pi Pico, and 74 ms on an ESP32. The inference time was with the same input data, a vector of size 144. To estimate the electrical charge consumption per inference (µAh), a Keweisi KWS-V20 USB Tester was used, and the inferences were made continuously, without delay, for 20 min. The total accumulated consumption is divided by the number of inferences made in that period, which is calculated based on the average inference time per device.

#### 3.3.2. Implementation

For the physical implementation part,, there are aspects to consider, such as where to obtain the month or season in which the inference is made, since it is another input to the network. The connection of the DHT22 sensor to the ESP32 is shown in Figure 10. To obtain this input variable, the ESP32 can be configured with a local Wi-Fi to obtain the date, by an RTC module, by using the millis function, or even by setting the embedding value to a specific month.

The network will function exactly the same as in the simulation under the same data; that is, a 144-value input vector will have the same output whether the inference is on a computer or an MCU. For the proper operation, an algorithm is required to sort the data and provide accurate updates at a given time.

As seen in the flowchart in Figure 11, setting a default month, we take every hour as an update, with a circular memory buffer; the number of buffers can be increased to measure it in real time for every minute or every 10, 15, or 20 min, where each buffer (independent of the other buffers) is an entry to the network, but the SRAM memory must be taken into consideration. Generally, each buffer occupies 576 bytes (144 values of 4 bytes, float type variables), so with six active buffers, the total consumption of only the buffers in RAM would be 3456 bytes. Once the buffer is filled or every time it is updated with a new value, it will have one output per buffer so that only at the beginning will the system require a 24-h calibration to begin predicting correctly.

### 3.4. Robustness Validation and Geographic Generalization

To evaluate the generalization capability of the proposed model, a geographic validation was performed using data from different cities around the world, selected to represent a wide variety of climates and latitudinal locations. In this evaluation, the performance of the SE-LSTM model was compared against a standard LSTM network, using the RMSE and the coefficient of determination R^2^ for temperature and humidity as the metrics.

The results are summarized in Table 8. In terms of temperature, SE-LSTM generally performs better, with a lower RMSE and a higher R^2^ in most cities. In terms of humidity, however, the performance is more variable. SE-LSTM achieves lower errors in some cases, while the classic LSTM achieves better R^2^ values in several cities.

## 4. Discussion

Using the different state-of-the-art architectures, the network parameters were optimized to maintain the best performance in this application, using the same dataset and test data to see how much the overall performance varies between them, in addition to the proposed model.

It was observed that the fixed embedding approach limits the adaptability to extreme weather events (e.g., El Niño), so it would be desirable to explore online learning mechanisms that adjust the vectors during the operation.

This network can still be optimized to achieve higher performance, but the limitation is that it would have to be able to operate in low-power embedded systems. As described in the implementation, six circular buffers consume approximately 3.4KB of SRAM, an amount manageable on an MCU.

Although TPA-LSTM demonstrated better performance in the design phase, its exclusion from the embedded evaluation was due to its incompatibility with lightweight deployment frameworks, such as TensorFlow Lite, used in edge computing environments. This technical limitation reinforces the need for optimized architectures such as SE-LSTM, which maintain good predictive performance while meeting the local execution, energy efficiency, and low latency requirements of embedded systems.

Future work could test variables other than the temperature and humidity, such as the pressure, wind direction, and speed. Applications could also be extrapolated to other fields, where only one variable per neural network model or multiple variables need to be predicted, requiring a local execution or short inference times.

However, the model performs optimally in contexts similar to those of the training; changing to a greenhouse or a very different biome requires retraining or readjusting the embeddings.

## 5. Conclusions

This research demonstrated that the use of embeddings in LSTM networks can be essential for improving climate projections in systems such as MCUs, reducing the network size without losing performance. This can be vital for fields such as precision agriculture or resource management, where issues such as latency, privacy, or energy efficiency are key components.

Although improvements in accuracy were noted, it was also evident that more complex models, such as TPA-LSTM, can offer an even better performance. However, the device where the inference will be performed, along with the environment where it may be monitored, are also taken into account. This solution offers an approach in remote scenarios such as rural, agricultural, or low-connectivity areas.

In tests on MCUs, SE-LSTM was shown to run in inference times between 55 ms and 671 ms, and its power consumption was lower compared to other lower-precision networks.

In short, while the study demonstrates progress in climate prediction, it opens the door to new research that could not only improve predictions but also have the potential to be applied in other areas, thanks to the power of edge computing and data processing. This is a powerful tool for a variety of future applications.

## Figures and Tables

**Figure 1 sensors-25-03601-f001:**
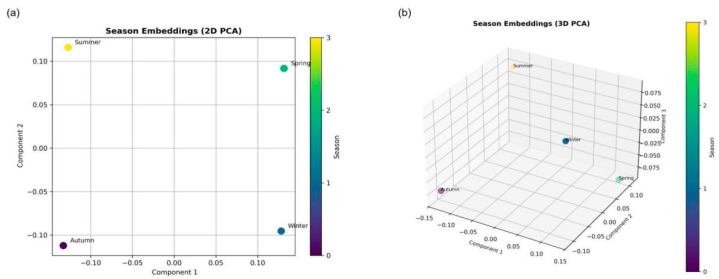
A PCA-based visualization of the 4D embeddings. (**a**) 2D projection. (**b**) 3D projection.

**Figure 2 sensors-25-03601-f002:**
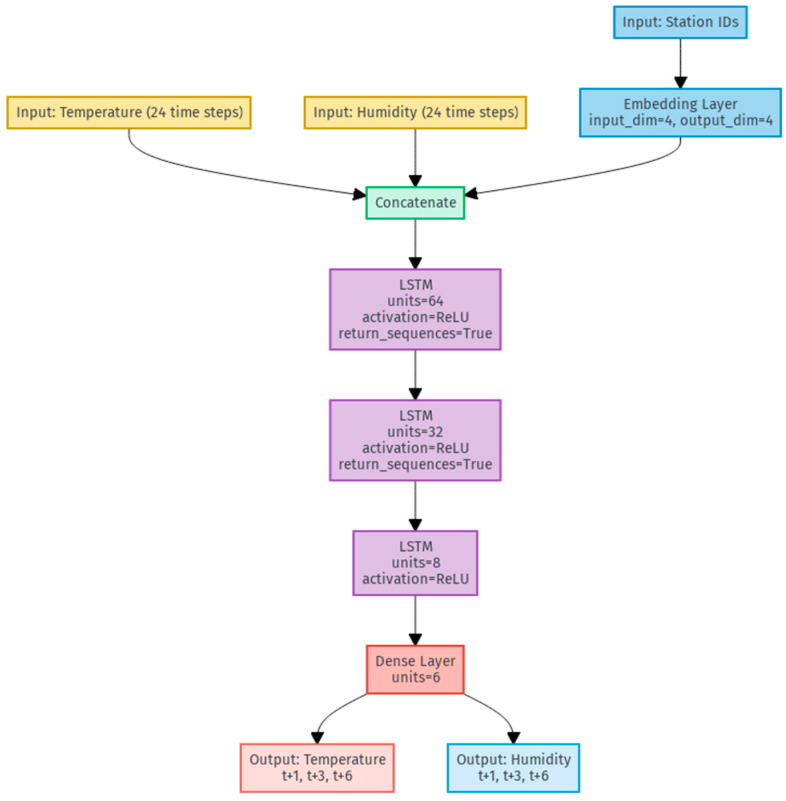
A flowchart of the model.

**Figure 3 sensors-25-03601-f003:**
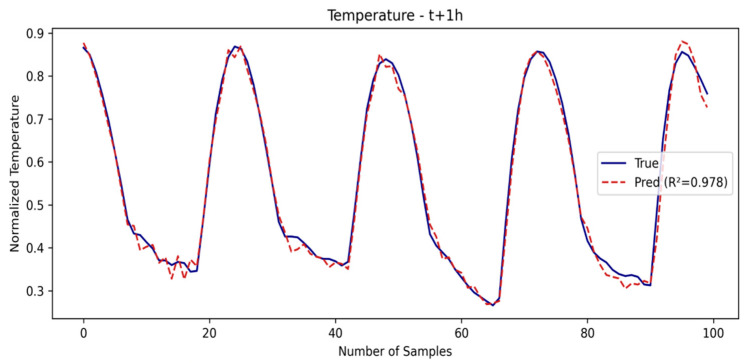
Temperature prediction in first window.

**Figure 4 sensors-25-03601-f004:**
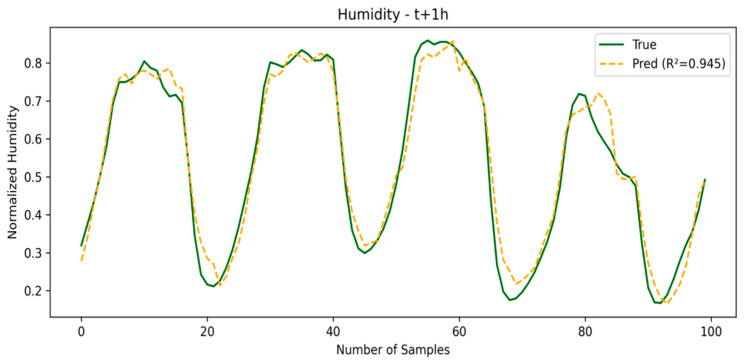
Humidity prediction in first window.

**Figure 5 sensors-25-03601-f005:**
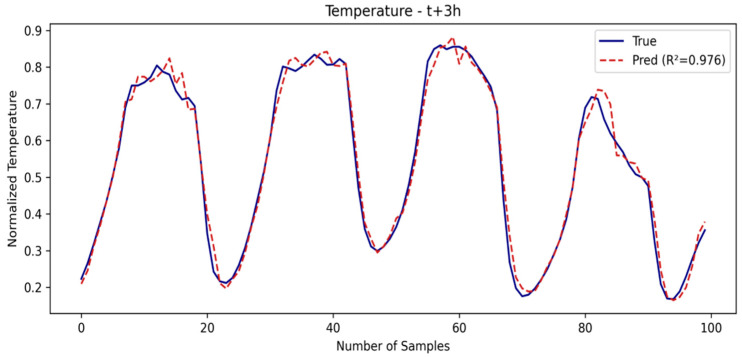
Temperature prediction in second window.

**Figure 6 sensors-25-03601-f006:**
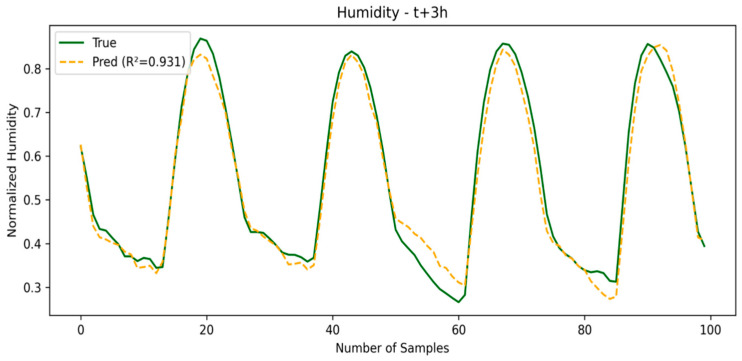
Humidity prediction in second window.

**Figure 7 sensors-25-03601-f007:**
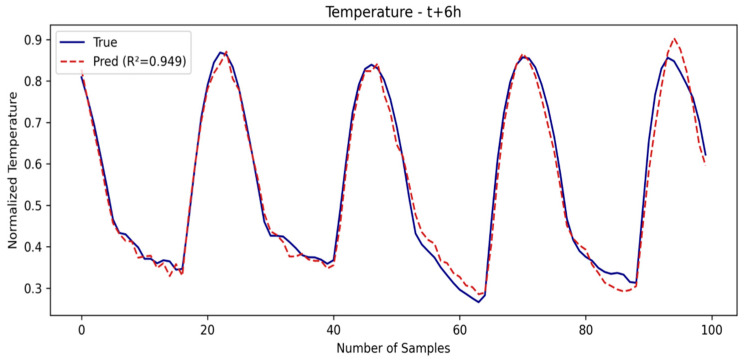
Temperature prediction in third window.

**Figure 8 sensors-25-03601-f008:**
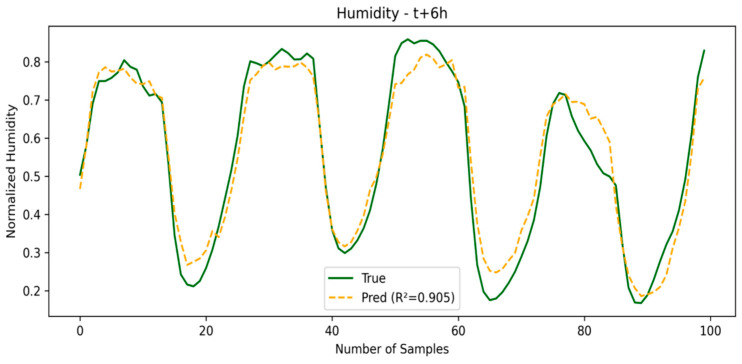
Humidity prediction in third window.

**Figure 9 sensors-25-03601-f009:**
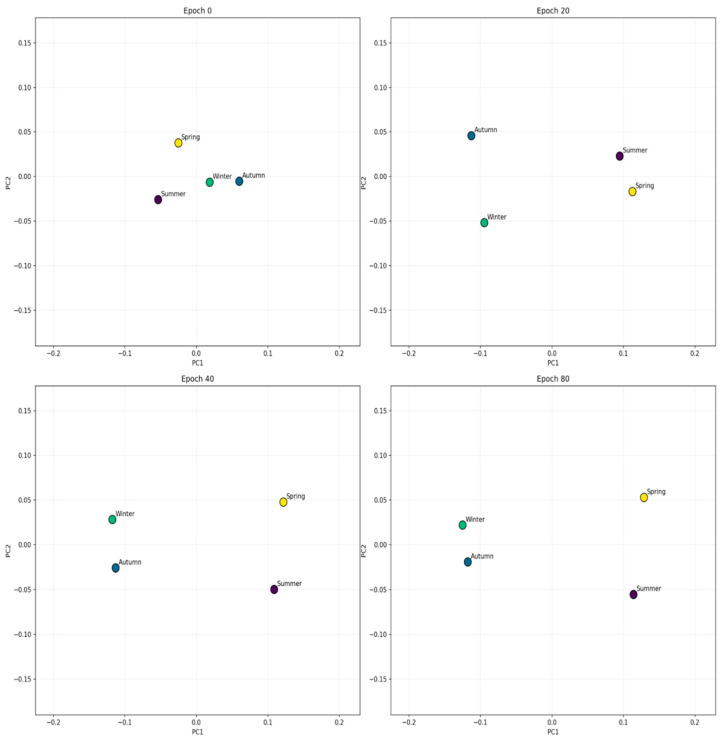
Evolution of embedding vectors during training.

**Figure 10 sensors-25-03601-f010:**
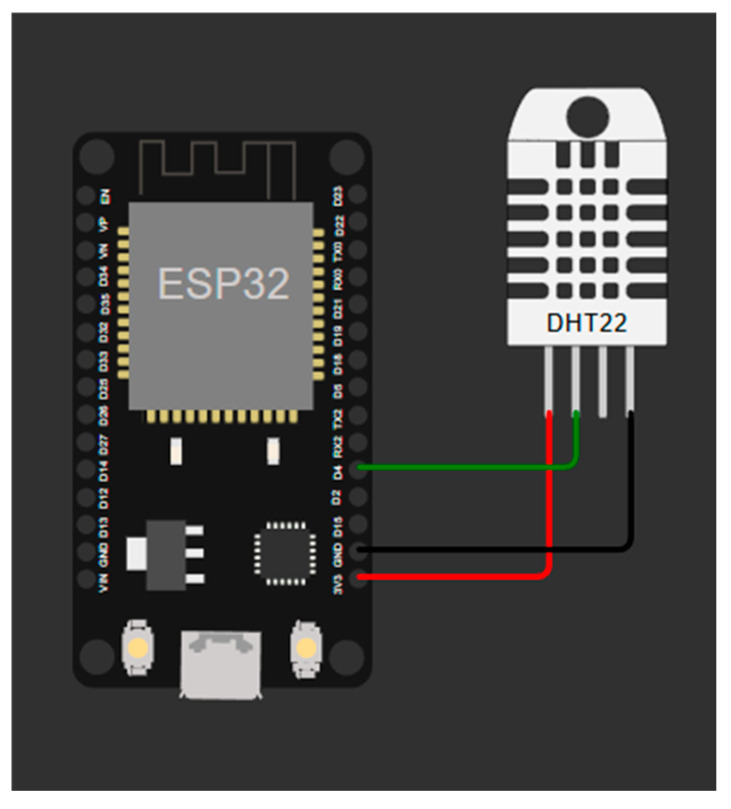
Connection of DHT22 sensor to ESP32 GPIO pin 4.

**Figure 11 sensors-25-03601-f011:**
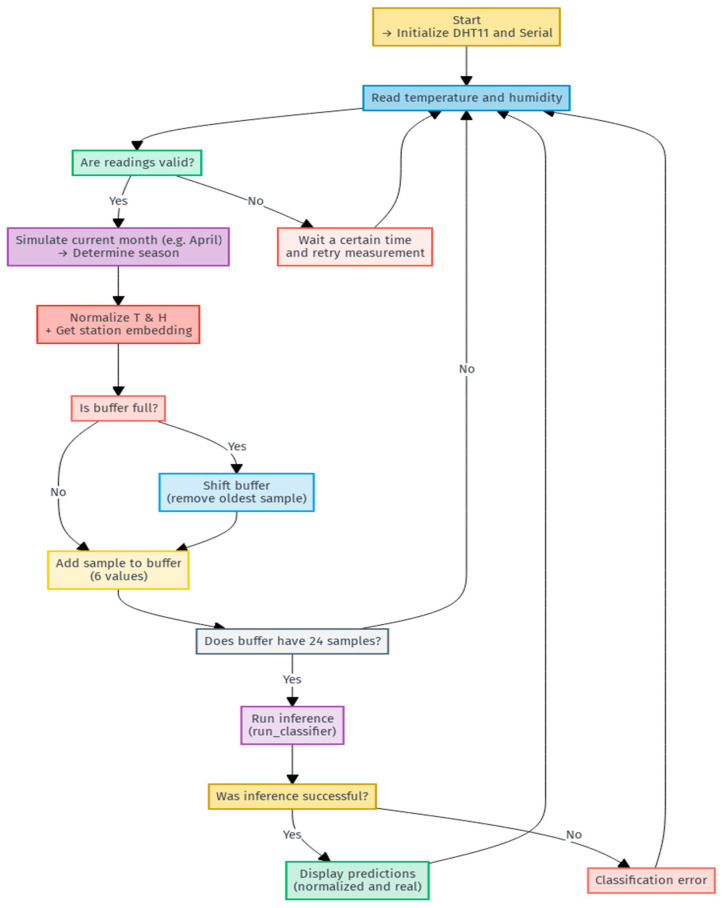
Operation flow for real application.

**Table 1 sensors-25-03601-t001:** First time window (t + 1).

Model	Variable	MSE	MAE	RMSE	MAPE	RSE	R^2^
SE-LSTM	Temperature	0.0005	0.0164	0.0234	3.48	0.0219	0.9781
	Humidity	0.0022	0.0349	0.0469	5.66	0.0551	0.9449
LSTM	Temperature	0.0005	0.0165	0.0228	3.48	0.0209	0.9791
	Humidity	0.0010	0.0226	0.0318	3.52	0.0252	0.9748
LMU	Temperature	0.0010	0.0224	0.0309	4.73	0.0381	0.9619
	Humidity	0.0015	0.0295	0.0384	4.54	0.0367	0.9633
TPA-LSTM	Temperature	0.0002	0.0108	0.0151	2.38	0.0091	0.9909
	Humidity	0.0004	0.0139	0.0194	2.12	0.0094	0.9906
CNN-LSTM	Temperature	0.0005	0.0152	0.0221	3.27	0.0196	0.9804
	Humidity	0.0009	0.0216	0.0304	3.34	0.0231	0.9769
TCN	Temperature	0.0005	0.0159	0.0224	3.37	0.0200	0.9800
	Humidity	0.0010	0.0224	0.0309	3.50	0.0238	0.9762

**Table 2 sensors-25-03601-t002:** Second time window (t + 3).

Model	Variable	MSE	MAE	RMSE	MAPE	RSE	R^2^
SE-LSTM	Temperature	0.0010	0.0224	0.0311	3.54	0.0242	0.9758
	Humidity	0.0017	0.0301	0.0413	6.29	0.0687	0.9313
LSTM	Temperature	0.0012	0.0250	0.0350	5.20	0.0492	0.9508
	Humidity	0.0024	0.0369	0.0495	5.83	0.0613	0.9387
LMU	Temperature	0.0015	0.0281	0.0384	5.83	0.0593	0.9407
	Humidity	0.0026	0.0386	0.0505	6.15	0.0639	0.9361
TPA-LSTM	Temperature	0.0010	0.0228	0.0324	4.81	0.0420	0.9580
	Humidity	0.0019	0.0315	0.0431	4.98	0.0463	0.9537
CNN-LSTM	Temperature	0.0013	0.0248	0.0355	5.20	0.0506	0.9494
	Humidity	0.0024	0.0361	0.0494	5.68	0.0611	0.9389
TCN	Temperature	0.0013	0.0257	0.0356	5.32	0.0511	0.9489
	Humidity	0.0023	0.0356	0.0478	5.62	0.0573	0.9427

**Table 3 sensors-25-03601-t003:** Third time window (t + 6).

Model	Variable	MSE	MAE	RMSE	MAPE	RSE	R^2^
SE-LSTM	Temperature	0.0013	0.0257	0.0358	5.28	0.0505	0.9495
	Humidity	0.0038	0.0476	0.0614	7.46	0.0949	0.9051
LSTM	Temperature	0.0017	0.0301	0.0412	6.26	0.0683	0.9317
	Humidity	0.0039	0.0478	0.0626	7.56	0.0986	0.9014
LMU	Temperature	0.0018	0.0309	0.0423	6.46	0.0722	0.9278
	Humidity	0.0036	0.0467	0.0604	7.42	0.0917	0.9083
TPA-LSTM	Temperature	0.0017	0.0292	0.0410	6.18	0.0679	0.9321
	Humidity	0.0037	0.0460	0.0609	7.13	0.0932	0.9068
CNN-LSTM	Temperature	0.0018	0.0298	0.0420	6.19	0.0711	0.9289
	Humidity	0.0039	0.0477	0.0627	7.50	0.0987	0.9013
TCN	Temperature	0.0017	0.0307	0.0418	6.37	0.0705	0.9295
	Humidity	0.0038	0.0470	0.0613	7.40	0.0944	0.9056

**Table 4 sensors-25-03601-t004:** Overall performance.

Model	Network Weight (KB)	MSE	MAE	RMSE	MAPE	RSE	R^2^
SE-LSTM	426	0.0017	0.0295	0.0399	5.29	0.0525	0.9475
LSTM	1114	0.0018	0.0298	0.0425	5.31	0.0406	0.9461
LMU	522	0.0020	0.0327	0.0445	5.86	0.0446	0.9398
TPA-LSTM	1460	0.0015	0.0257	0.0353	4.60	0.0447	0.9554
CNN-LSTM	856	0.0018	0.0292	0.0404	5.19	0.0540	0.9460
TCN	785	0.0018	0.0296	0.0419	5.27	0.0394	0.9472

**Table 5 sensors-25-03601-t005:** Vectors obtained after training.

Season	x1	x2	x3	x4
Winter	−0.07277188	0.01109327	−0.09172057	−0.08308954
Spring	−0.05201762	−0.11797583	0.06664423	0.03551533
Summer	0.08841707	−0.01442201	0.06266572	−0.04614694
Autumn	−0.04601293	0.10943006	0.0083359	0.05554458

**Table 6 sensors-25-03601-t006:** Performance by prediction window.

Window	Model	Variable	MSE	MAE	RMSE	MAPE	RSE	R^2^
t + 1	SE-LSTM	Temperature	0.0005	0.0158	0.0226	3.34	0.0205	0.9795
		Humidity	0.0024	0.0361	0.0487	5.69	0.0593	0.9407
	LSTM	Temperature	0.0006	0.0164	0.0240	3.55	0.0230	0.9770
		Humidity	0.0025	0.0371	0.0501	5.85	0.0630	0.9370
	TCN	Temperature	0.0006	0.0171	0.0236	3.61	0.0223	0.9777
		Humidity	0.0011	0.0252	0.0333	3.80	0.0277	0.9723
	CNN-LSTM	Temperature	0.0005	0.0162	0.0227	3.39	0.0205	0.9795
		Humidity	0.0009	0.0221	0.0308	3.43	0.0236	0.9764
t + 3	SE-LSTM	Temperature	0.0010	0.0221	0.0309	3.43	0.0237	0.9763
		Humidity	0.0017	0.0296	0.0409	6.16	0.0675	0.9325
	LSTM	Temperature	0.0010	0.0227	0.0314	3.51	0.0245	0.9755
		Humidity	0.0018	0.0308	0.0430	6.54	0.0744	0.9256
	TCN	Temperature	0.0013	0.0261	0.0365	5.36	0.0534	0.9466
		Humidity	0.0025	0.0374	0.0497	5.83	0.0618	0.9382
	CNN-LSTM	Temperature	0.0012	0.0253	0.0352	5.19	0.0498	0.9502
		Humidity	0.0024	0.0360	0.0487	5.68	0.0595	0.9405
t + 6	SE-LSTM	Temperature	0.0013	0.0256	0.0356	5.31	0.0508	0.9492
		Humidity	0.0037	0.0470	0.0610	7.43	0.0936	0.9064
	LSTM	Temperature	0.0013	0.0256	0.0360	5.41	0.0522	0.9478
		Humidity	0.0044	0.0511	0.0666	8.00	0.1117	0.8883
	TCN	Temperature	0.0018	0.0299	0.0420	6.18	0.0711	0.9289
		Humidity	0.0038	0.0475	0.0619	7.37	0.0962	0.9038
	CNN-LSTM	Temperature	0.0018	0.0315	0.0427	6.50	0.0734	0.9266
		Humidity	0.0039	0.0480	0.0628	7.59	0.0991	0.9009
Average general	SE-LSTM	-	0.0017	0.0294	0.0399	5.23	0.0526	0.9472
	LSTM	-	0.0019	0.0306	0.0419	5.48	0.0581	0.9419
	TCN	-	0.0018	0.0305	0.0429	5.36	0.0415	0.9446
	CNN-LSTM	-	0.0018	0.0299	0.0405	5.30	0.0543	0.9457

**Table 7 sensors-25-03601-t007:** Inference time and current cost of various MCUs with SE-LSTM, TCN, CNN-LSTM, and LSTM.

Model/Peak RAM Usage	MCUs	Processor	Inference Time	Current Cost Per Inference
SE-LSTM	Raspberry Pi Pico	RP2040	671 ms	41.90 µAh
85.8 KB	ESP32	Xtensa LX6	74 ms	5.12 µAh
	ESP32-S3	Xtensa LX7	55 ms	3.79 µAh
TCN	Raspberry Pi Pico	RP2040	1804 ms	112.7 µAh
43.6 KB	ESP32	Xtensa LX6	210 ms	14.47 µAh
	ESP32-S3	Xtensa LX7	171 ms	11.87 µAh
CNN-LSTM	Raspberry Pi Pico	RP2040	1638 ms	102.38 µAh
30.0 KB	ESP32	Xtensa LX6	202 ms	13.62 µAh
	ESP32-S3	Xtensa LX7	152 ms	10.55 µAh
LSTM	Raspberry Pi Pico	RP2040	644 ms	40.25 µAh
85.7 KB	ESP32	Xtensa LX6	73 ms	5.05 µAh
	ESP32-S3	Xtensa LX7	54 ms	3.69 µAh

**Table 8 sensors-25-03601-t008:** A comparison of the performance of the SE-LSTM and LSTM models in the prediction of temperature and humidity in different cities around the world.

City	Model	RMSE Temperature	RMSE Humidity	R^2^ Temperature	R^2^ Humidity
Longyearbyen	SE-LSTM	0.0261	0.0659	0.9510	0.6768
	LSTM	0.0298	0.0690	0.9465	0.6687
Oslo	SE-LSTM	0.0220	0.0333	0.8937	0.6325
	LSTM	0.0303	0.0270	0.8853	0.6373
Madrid	SE-LSTM	0.0329	0.0701	0.9106	0.7791
	LSTM	0.0348	0.0821	0.8138	0.8053
Beijing	SE-LSTM	0.0319	0.0698	0.9380	0.8263
	LSTM	0.0285	0.0809	0.8952	0.8387
Cairo	SE-LSTM	0.0295	0.0635	0.9645	0.9132
	LSTM	0.0261	0.0717	0.9502	0.9240
Cancún	SE-LSTM	0.0273	0.0507	0.9546	0.8767
	LSTM	0.0320	0.0484	0.9271	0.9073
Manila	SE-LSTM	0.0202	0.0365	0.9717	0.9269
	LSTM	0.0204	0.0367	0.9708	0.9262
San José	SE-LSTM	0.0249	0.0398	0.9730	0.9356
	LSTM	0.0298	0.0391	0.9600	0.9414
Quito	SE-LSTM	0.0308	0.0487	0.9666	0.9358
	LSTM	0.0359	0.0452	0.9484	0.9551
Piura	SE-LSTM	0.0224	0.0379	0.9873	0.9689
	LSTM	0.0279	0.0363	0.9795	0.9769
East London	SE-LSTM	0.0210	0.0443	0.9363	0.7563
	LSTM	0.0298	0.0397	0.8666	0.8243
Sidney	SE-LSTM	0.0290	0.0626	0.9067	0.7542
	LSTM	0.0307	0.0636	0.8475	0.8276
Invercargill	SE-LSTM	0.0335	0.0751	0.9185	0.7295
	LSTM	0.0406	0.0787	0.8265	0.7903

## Data Availability

All resources used in this research (dataset, code, and trained models) are available in the GitHub repository and Google Drive: https://github.com/JhanSudoAPT/SE-LSTM-MCU (accessed on 9 May 2025); https://drive.google.com/drive/folders/1NegTTShIXsTYjQVaiSDWSIg_QDC5_z5k?usp (accessed on 9 May 2025).

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
