# Peer review of "Design and Implementation of an LSTM Model with Embeddings on MCUs for Prediction of Meteorological Variables"

_sensors, 2025, doi:10.3390/s25123601_

Round 1

Reviewer 1 Report

Comments and Suggestions for Authors

The paper proposes a novel LSTM variant called SE-LSTM (Single Embedding LSTM) for meteorological variable prediction on resource-constrained MCUs. The model integrates embedding layers to capture seasonal patterns in temperature and humidity data, aiming to reduce model size while maintaining prediction accuracy.

The authors are encouraged to address the following comments to improve the technical depth of their paper and its scientific thoroughness.

The 4D seasonal embeddings are trained on Peruvian climate data (-16.4333°S, -71.5617°W). Their effectiveness in other climatic zones (e.g., equatorial or polar) remains unverified.

Benchmarking excludes memory usage analysis during multi-buffer operation (critical for real-time 10-minute predictions), and no power consumption metrics are provided despite emphasizing low-power design.

The static embedding approach cannot adapt to climate anomalies (e.g., El Nino), unlike adaptive methods such as online learning LSTMs.

While TPA-LSTM shows superior accuracy, its exclusion from MCU implementation comparison, e.g., Grieco, Luigi Alfredo, et al., eds. Ad-Hoc, Mobile, and Wireless Networks: 19th International Conference on Ad-Hoc Networks and Wireless, ADHOC-NOW 2020, Bari, Italy, October 19–21, 2020, Proceedings. Vol. 12338. Springer Nature, 2020 (due to TFLite incompatibility) weakens the edge computing claims, thus the provided discussion of the related work needs to be updated.

The model processes only temperature/humidity, ignoring potentially synergistic variables like solar radiation or wind speed mentioned in the discussion.

How does SE-LSTM compare to temporal convolutional networks (TCNs) in terms of MCU resource usage and prediction accuracy?

Reviewer 2 Report

Comments and Suggestions for Authors

The article is a significant contribution to the field of peripheral computing and time series forecasting. The proposed SE-LSTM model demonstrates a good balance between accuracy and resource efficiency, which makes it applicable in real IoT systems. However, to improve the quality of work, it is necessary to eliminate the following disadvantages:

  1. It should be explained exactly how the missing values (-999) were processed and what methods were used to fill them in.
  2. In the section on embedding in microcontrollers, it is necessary to specify exactly how seasonality is handled, for example, how the current season for embedding is determined. 
  3. It is recommended to add an analysis of the model's energy consumption on different devices, as this is a key parameter for IoT solutions.
  4. It is recommended to provide comparisons with more modern architectures such as Transformer or Hybrid CNN-LSTM, which may show better performance.
  5. Graphs (for example, PCA for seasonality) are presented without detailed analysis. It is important to explain exactly how the embedding vectors reflect seasonal patterns.
  6. It is recommended to check the design, whether indents are needed between paragraphs.
  7. The distance between the paragraph "The following figures show how the seasons are related, represented by vectors in 99 of the 2D and 3D planes:" and Figure 1 is too large.
  8. It is recommended to arrange Figures 1 and 2 in such a way that there is not such a large empty distance between them (for example, to change their size or position).
  9. In section 3, "Results", the subsections should be renumbered. For example, the subsections "Results and comparison" and "Comparison at the design stage" have the same numbers – 3.1. This problem can be solved simply by deleting the heading "3.1. Results and comparison".
  10. In section 3, "Results", most paragraphs are missing indents.
  11. The captions of the figures should be brought into a single form, since somewhere there is a dot (for example, Figure 1:), and somewhere there is a colon (for example, Figure 11.). There is also a dot at the end of some picture captions (for example, Figure 1: 2D PCA plot illustrating how seasons cluster in the embedded space.), and in some there is not (for example, Figure 2: 3D PCA graph).
  12. The caption to table 6 is located at the bottom of the table.
  13. The table captions should be unified. Table 7 has a dot at the end of the signature (Table 7. Inference time on various MCUs.), but there are no other tables with dots.

Round 2

Reviewer 1 Report

Comments and Suggestions for Authors

The authors have addressed my comments. 

Reviewer 2 Report

Comments and Suggestions for Authors

The authors have taken into account all the comments, the article does not require any additional edits.